# The Effect of Core Stabilization Exercise with the Abdominal Drawing-in Maneuver Technique on Stature Change during Prolonged Sitting in Sedentary Workers with Chronic Low Back Pain

**DOI:** 10.3390/ijerph19031904

**Published:** 2022-02-08

**Authors:** Pongsatorn Saiklang, Rungthip Puntumetakul, Thiwaphon Chatprem

**Affiliations:** 1Division of Physical Therapy, Faculty of Physical Therapy, Srinakharinwiroj University, Nakhonnayok 26120, Thailand; Pongsatornsa@g.swu.ac.th; 2Research Center of Back, Neck, Other Joint Pain and Human Performance (BNOJPH), Khon Kaen University, Khon Kaen 40002, Thailand; Thiwaphon.ao@gmail.com; 3Department of Physical Therapy, Faculty of Associated Medical Sciences, Khon Kaen University, Khon Kaen 40002, Thailand

**Keywords:** spinal load, core stability, ergonomics

## Abstract

To enhance stature recovery, lumbar spine stabilization by stimulating the deep trunk muscle activation for compensation forces originating from the upper body was introduced. The abdominal drawing-in maneuver (ADIM) technique has been found mainly to activate deep trunk muscles. The purpose of the current study was to determine whether 5 weeks of training of deep trunk muscles using the ADIM technique could improve stature recovery, delay trunk muscle fatigue, and decrease pain intensity during prolonged sitting. Thirty participants with chronic low back pain (CLBP) conducted a core stabilization exercise (CSE) with the ADIM technique for 5 weeks. Participants were required to sit for 41 min before and after the exercise intervention. Stature change was measured using a seated stadiometer with a resolution of ±0.006 mm. During sitting, the stature change, pain intensity, and trunk muscle fatigue were recorded. A comparison between measurements at baseline and after 5 weeks of training demonstrated: (i) stature recovery and pain intensity significantly improved throughout the 41 min sitting condition; (ii) the bilaterally trunk muscle showed significantly decreased fatigue. The CSE with the ADIM technique was shown to provide a protective effect on detrimental reductions in stature change and trunk muscle fatigue during prolonged sitting in young participants under controlled conditions in a laboratory. This information may help to prevent the risk of LBP from prolonged sitting activities in real life situations.

## 1. Introduction

Sedentary behavior is characterized by an energy expenditure less than or equal to 1.5 metabolic equivalents (METs) while in a sitting or reclining position when awake [1]. Today, increasing numbers of individuals spend extended periods in a seated position at work as well as during leisure time [2,3]. Recently, sedentary workers in Thailand reported recurring low back pain, with 63% showing that their low back pain was aggravated by sitting during working hours [4]. Chronic low back pain (CLBP) has a global prevalence of 11–23% among people with low back pain [5,6]. The socioeconomic burden of CLBP stems from a prolonged loss of function, which consequently results in decreased work productivity and increased medical costs [5,7].

Deep trunk muscle fatigue may arise from continuous contraction of the trunk muscles during prolonged sitting [8,9,10]. This fatigue reduces muscular support to the spine and increases stress on ligaments and intervertebral discs [9,10]; consequently, it reduces intervertebral disc height [11,12]. Reductions in disc height could increase compressive stress on sensitive spinal structures [13,14] and may stimulate nociceptor activity, leading to pain [14]. Stature change measurement is a method used to reflect alterations in spinal length, and the reduction of spinal length is known as spinal shrinkage or stature loss [15]. Prolonged sitting postures could lead to stature reduction and ultimately to low back pain [9,16,17]. 

Trunk muscles play an essential role in contributing to spinal stability [18]. There are two types of trunk muscle systems: superficial and deep [19,20]. The internal oblique (IO), transversus abdominis (TrA), and lumbar multifidus (LM) muscles represent a deep muscle system that compensates for forces on the upper body of the spine and increases lumbar stability [18,21]. Previous studies reported changes in the muscle recruitment pattern and timing of muscle onset in people with low back pain [22,23]. Increased superficial trunk muscle activation occurs to compensate for deep trunk muscle dysfunction [24,25], in which the neural control subsystem attempts to maintain spinal stability [18,26]. Increased activation of the superficial trunk muscle can compress the spinal structure and lead to delayed stature recovery [25,27].

Previous research reported that superficial trunk muscle activity can be reduced by activating the deep trunk muscles using the abdominal drawing-in maneuver (ADIM) technique [15,28,29]. The ADIM technique is known to elicit preferential recruitment of the transversus abdominis muscle with minimal activation of the superficial trunk muscles. For this technique, participants are instructed to ‘gently draw in their lower abdomen toward the spine’ [13]. Saiklang et al. (2020) investigated the change in stature recovery in patients with CLBP following the immediate effect of the ADIM technique for 1 min repeated three times throughout a 41 min prolonged sitting period. The results demonstrated that the ADIM technique significantly improved stature recovery and increased TrA and IO muscle activities and TrA and IO/RA ratios compared with upright sitting without exercise [15]. 

To date, no investigation has reported the effect of the long-term impact of the core stabilization exercise (CSE) with the ADIM technique program focusing on deep trunk muscle on stature recovery during prolonged sitting. The CSE with ADIM technique aims to improve neuromuscular control skills, relearn normal function, and enhance endurance of the deep muscles around the lumbar spine, such as the TrA and LM muscles [28,30]. 

The aim of the current study was to investigate differences in stature change, pain, and trunk muscle fatigue during prolonged sitting in seated sedentary workers with CLBP between baseline and the first day after a 5-week CSE with the ADIM technique. We hypothesized that the 5-week CSE with the ADIM technique can improve deep trunk muscle endurance, reduce pain, and delay stature reduction during prolonged sitting.

## 2. Materials and Methods

### 2.1. Design and Setting

The study used a within-subject repeated-measures design. It was conducted at the research center of the Back, Neck, Other Joint Pain and Human Performance (BNOJPH) laboratory, Khon Kaen University, Thailand. Ethics approval for this research was granted before the study by the Human Research Ethics Committee (HE612220) of Khon Kaen University. The study was registered at clinicaltrials.in.th (registration number: TCTR20180823004).

### 2.2. Participants 

Thirty participants, aged 20–39 years, were recruited via posters on bulletin boards at Khon Kaen University. Fifteen males and fifteen females were recruited to reduce the influence of gender. Inclusion criteria for the participants were established as follows: CLBP lasting more than three months, mild to moderate levels of pain on the numerical rating scale (NRS; ≤7 score) [31,32], low levels of activity limitation on the Roland Morris disability questionnaire (RMDQ; ≤12 score) [33], and reported sitting for at least two hours on any working day [9]. Participants were excluded if they had previous vertebral surgery, had been identified with a medical condition that affected spinal soft tissues, or were pregnant [32,34].

### 2.3. Sample Size Determination 

The sample size was calculated after preliminary data collection from 12 participants (six male and six female) who performed the CSE with ADIM for 5 weeks. The mean difference of the stature changes before and after the exercise intervention was set at 3 mm. A significance level of 0.05 (Zα (0.05) = 1.96) and a power of 90% (Zβ (0.1) = 1.28) were used in the calculation. After an additional 15% correction for dropouts, the number of subjects was 21. Thus, the current study required at least 30 participants (15 males and 15 females for balanced gender) to achieve sufficient statistical power for the analyses.

### 2.4. Outcome Measurements

#### 2.4.1. Stature Change Response

Stature change response was measured using a seated stadiometer device (certified Thai petty-patent No. 5607; Figure 1) [15,34]. The Digimatic Indicator identifies variations in stature change with a resolution of ±0.006 mm and was used to measure stature change. The device displays real-time data and repeatedly records data up to 5 Hz (ID-C 150, 1050 Digimatic Indicator, Manual No. 3061, Series No. 543, Mitutoyo, Kawasaki, Japan). 

The position of the participants was controlled during the measurements as follows: 

(i) The Digimatic Indicator settings allowed the distal end to rest directly on the highest apex of the skull to ensure its consistent positioning throughout participant repositioning (Figure 1) [15,34].

(ii) Head positioning with the eyes kept level was maintained by coaching the participants to concentrate on a visual cue, a letter on an alphabet chart, placed at eye level on the opposite end (Figure 1B) [15,34].

(iii) The wooden seat platform and footrest were adjustable so that the participants’ ankle, knee, and hip joints were positioned at 90° throughout the measurements. Heels touched the back of the footrest [35]. The head and sacral supports were adjustable to accommodate the participants’ spinal posture [35]. A pillow was placed on the participants’ lap to support their forearms positioned at 90° to their upper arms (Figure 1A).

(iv) The spinal alignment was controlled by sensors placed on the spinous processes of the following vertebrae: cervical spine 4, thoracic spine 4, thoracic spine 12, and lumbar spine 3. These sensors were connected to a light diode feedback located opposite the seated participant (Figure 1B) [12,15]. These sensors were used as a measure for control to ensure that the participants maintained the same posture throughout the experiment period. The process of stature change measurement was performed by Researcher P.S.

#### 2.4.2. Trunk Muscle Fatigue

Eight pairs of Ag-AgCl disposable surface electromyography (sEMG) electrodes (EL 503) with electrical contact surface areas of 1 cm^2^ and a center-to-center spacing of 2.5 cm were attached parallel to each muscle on both sides: to the rectus abdominis muscle (RA) [36], the internal oblique and transversus abdominis (IO and TrA) [37], the iliocostalis lumborum pars thoracis (ICLT) [38], and the lumbar multifidus (LM) [30] after skin abrasion and cleaning with alcohol. Electromyography (EMG) data were recorded at 2000 Hz using the Wireless Bipolar Cometa Mini Wave Plus 16-channel EMG system (Cometa, Bareggio, Italy), an online band-pass filter (10–500 Hz), and a 60 Hz notch filter (power line in Thailand). The raw EMG signal was first visually checked for electrocardiac artifacts. The raw EMG signal was processed with fast Fourier transformation to determine the median frequency (MDF) value (Hz). The decrease in the MDF of the EMG signal was taken as an indirect measure of muscle fatigue [6]. Trunk muscle fatigue was collected by Researcher P.S.

#### 2.4.3. Pain Rating Scale

Pain intensity was assessed using an 11-point numerical rating scale (0–10 NRS). Subjective measures of pain were obtained from the NRS, employed to assess pain on a scale ranging from 0 (no pain) to 10 (worst possible pain) [31,39]. This outcome measurement was evaluated by Researcher T.C.

#### 2.4.4. Functional Disability

The Roland Morris disability questionnaire (RMDQ) Thai version was used to assess functional disability due to low back pain [40]. This questionnaire includes 24 items [41], which were rated by Researcher T.C.

### 2.5. Procedure 

The flowchart of the current study is presented in Figure 2. Thirty-three participants were recruited from the advertisements. After the screening process, 30 participants were included in the study. Three participants were excluded due to experiencing low back pain >7 based on the NRS. Thirty participants meeting the inclusion criteria were asked to visit the research laboratory.

On the first day, the participants were familiarized with the stature change measurement. These involved participants practiced stepping in and out of the stadiometer until a standard deviation (SD) of <0.5 mm was achieved over ten repeated stature measurements [12,42]. Then, they were asked about their disability score (RMDQ) and pain intensity at rest (NRS). 

On the second day, all participants arrived within an hour of waking, between 8 and 10 a.m. [42,43] to avoid stature loss before the test trial. They were requested to sleep for at least 8 h each night before the days of the experiment [44]. They were asked to undertake normal activities of daily living, refrain from vigorous physical activities, and refrain from alcohol consumption for 24 h before the experimental sessions [45]. After the application of surface electrodes, the participants were asked to maintain the Fowler’s position (lying) for 20 min to eliminate any abnormal spinal loading that may have been present before arrival [34,45]. Then, they were asked to sit for 41 min, and the outcome measurements were collected, as shown in Figure 3.

Next, the participants were asked to practice CSE using the ADIM technique with the researcher R.P. When they could perform this correctly, they were required to exercise CSE with ADIM for 5 weeks. After 5 weeks of training, all participants were asked to stop their exercises completely. The outcomes were re-measured on the first day after the 5-week CSE with the ADIM technique. 

### 2.6. Prolonged Sitting Condition 

An overview of the prolonged sitting condition, with time points and their outcome measurements, is shown in Figure 3. Participants sat in the seated stadiometer, according to conditions described in 2.4.1, with the Digimatic Indicator in contact with the skull apex, marked by a waterproof pen. During the measurements, the participants remained in the same posture without speaking. To reduce errors in the spinal change measurements due to involuntary movements and slight differences in the breathing phase, all measurements were taken at the end of the expiration phase of the breathing cycle [35,46]. Each measurement set, consisting of 75 data points sampled over 15 s, was considered at time 0 and at the end of a 2 min interval, which reduced the effect of variations in the stature change assessment due to both breathing patterns and uncontrolled movements [12,34,35].

A baseline stature measurement set was recorded (T_0_). During the test trials, the participants remained in a freestyle sitting position, which did not require a straight back, without a backrest for 10 min. Then, the stature change (Tsit) in the participants was measured to be used as a normalized value. Next, the participant was asked to sit upright for 1 min. The stature change and pain intensity were measured at the end of each session (T1 [at 13–15 min], T2 [at 26–28 min], and T3 [at 39–41 min]). The raw sEMG signal was processed using the triangle-Bartlett method of fast Fourier transformation to determine the median frequency (MDF) value. The sEMG data were retrieved every 10 min block of sEMG data from the 41 min sitting period (at 0–10, 15–25, and 28–38 min) for analysis, Figure 3. The total time for each test trial was 41 min. Participants were not allowed to stand during the test trials.

### 2.7. Core Stabilization Exercise (CSE) with ADIM Technique

The exercise program was supervised by a physical therapist with 30 years of experience (RP). This exercise program was modified from Puntumetakul et al. [28]. The details of the CSE each week are appended (Table A1). Researcher R.P. trained this exercise to all participants face-to-face in a 20 min session. The participants were re-assessed with researcher R.P. twice a week for 5 weeks at the laboratory to determine whether they could successfully perform the previous exercise. As the CSE with the ADIM technique was a milestone exercise, if the participants failed to perform the previous exercise accurately, they were retrained in the previous exercise until they succeeded. The participants were required to perform a daily set of home exercises of the same level, position, and frequency as those demonstrated during the exercise session with the physical therapist. The participants were asked to record in their logbook a daily home exercise program, including position, duration, and frequency of the exercise, as well as a record of their drug and alternative treatment throughout the study period and any adverse effects of the exercise. During the exercise, one of the researchers contacted the participants by telephone every week to motivate them to continue their daily home exercises. After 5 weeks of training, all participants were asked to stop their exercises completely.

### 2.8. Data Analysis

The mean and standard deviation (SD) were used to assess participants’ demographics and data of stature change at each time of measurement and were calculated from the reference point of T_sit_. Differences in stature within a condition were assessed using a one-way repeated measure ANOVA for time effect (T_1_, T_2_, and T_3_) with the Bonferroni post-hoc analysis (significant at *p* < 0.017; 0.05/3).

The differences in trunk muscle fatigue and pain intensity within the condition for non-normally distributed data were analyzed using the Friedman test, and post hoc tests were conducted using Wilcoxon signed-rank tests. A significance level was set at *p* < 0.05 for trunk muscle fatigue and pain intensity.

Data comparisons before and after the first day of the 5-week CSE regarding pain at rest and functional disability were analyzed using the paired *t*-test (*p* < 0.05). During the prolonged sitting condition, the data comparison before and after the first day of 5 weeks of the exercise on stature change was analyzed using the paired *t*-test (*p* < 0.05). Further, the pain intensity and trunk muscle fatigue were analyzed using Wilcoxon signed-rank tests (*p* < 0.05). 

All analyses were performed using SPSS version 19.0 software (SPSS Inc., Chicago, IL, USA). The Shapiro–Wilk test was performed to check the data distribution.

## 3. Results

### 3.1. Participant Characteristics

All participants achieved the preferred level of repeatability for the stature change measurements (SD ≤ 0.5 mm). The participants reported that they did not use any drug for reducing their low back pain and had no adverse repercussions of the exercise throughout the 5 weeks of the training. The demographic data and clinical characteristics are presented in Table 1.

### 3.2. Pain Intensity at Rest and Functional Disability

The results showed a significant reduction in pain intensity (mean difference: 2.14 ± 1.50 (95% CI: 1.57 to 2.69) at *p* < 0.001) and functional disability (mean difference: 2.33 ± 1.81 (95% CI: 1.66 to 3.01) at *p* < 0.001) between baseline and the 5-week CSE with the ADIM, as shown in Figure 4 and Figure 5, respectively.

### 3.3. Stature Changes during Sitting

The stature changes during sitting before and after CSE are shown in Table 2. The result of stature change after sitting for 10 min (T_sit_) showed no significant differences between baseline and after the 5-week CSE with the ADIM technique (*p* = 0.458). This result indicates that T_sit_ between baseline and after the 5-week CSE with the ADIM technique was comparable and could be used as a reference point for the stature changes calculation at T_1_, T_2_, and T_3_.

At baseline, the results of the current study illustrated that the baseline showed a significant reduction in stature due to time (T_1_, T_2_, and T_3_) (*p* < 0.017). In the same pattern, the results illustrated that after the 5-week CSE with the ADIM technique, there was a significant reduction in stature due to time.

Comparing baseline and the 5-week CSE with the ADIM technique, the stature changes at T_1_, T_2_, and T_3_ were significantly improved in the 5-week CSE with the ADIM technique (Table 2).

### 3.4. Pain Intensity during Sitting

At baseline, the pain intensity of T_2_ and T_3_ were significantly increased compared to other time of measurements (T_sit_ and T_1_). After the 5-week CSE with the ADIM technique, the pain intensity of T_2_ (*p* < 0.05) and T3 (*p* < 0.001) was significantly increased from T_sit_. At T_3_, pain intensity was significantly increased from T_1_ and T_2_ (*p* < 0.05). 

Comparing baseline and the 5-week CSE with the ADIM technique, the pain intensity at T_1_, T_2_, and T_3_ was significantly decreased in the 5-week CSE with the ADIM technique, as shown in Table 3.

### 3.5. Trunk Muscle Fatigue during the Experiment

At baseline, the MDF in the sitting condition is shown in Table 4. The Friedman test revealed a significant difference in the MDF values in the muscles and both sides at each time of measurement during prolonged sitting. The Wilcoxon signed-rank tests showed a significant difference between the measurement times. For both sides of the TrA and IO muscles, the MDF value at the 15th–25th min was significantly decreased compared to that of the 0–10th min. A further reduction in the MDF value was observed at the 28th–38th min.

After the 5-week CSE with the ADIM technique, the Friedman test did not reveal a significant difference in the MDF value in trunk muscles and both sides at all times of measurement. Compared with baseline values, the 5-week CSE with the ADIM technique showed a significant improvement in the MDF values (both sides of TrA and IO, the MDF value of 15th–25th and 28th–38th min).

## 4. Discussion

The aim of this study was to investigate differences in stature change, pain, and trunk muscle fatigue during prolonged sitting conditions in sedentary workers with CLBP between baseline and the 1st day after the 5-week CSE with the ADIM technique.

The results of the current study showed that the 5-week CSE with the ADIM technique provided a significant decrease in resting pain (mean difference: 2.14 ± 1.50; *p* < 0.001) and improvement in functional disability (mean difference: 2.33 ± 1.81; *p* < 0.001). The result of the current study agreed with those of previous studies that reported the potential of CSE to improve functional disability in patients with CLBP [47,48,49]. The results of the current study demonstrated that the CSE program might be clinically advantageous for CLBP patients with functional disability improvement by reducing pain.

During prolonged sitting, forces from bodyweight cause deformation of the elastic components of the disc and increased intra-discal pressure [50,51]. Fluid loss is known to occur when the pressure inside the disc increases and can be indicated as the major mechanism to account for the reduction in disc height and consequent stature loss [52,53]. Prolonged sitting in CLBP participants induced stature loss (mean difference −7.365 mm) at 41 min (T_3_) (Table 2).

The current study revealed that bilateral TrA and IO muscle fatigue occurred earlier during sitting (approximately 15–25 min after sitting) (Table 3). Sitting for prolonged periods has been partly attributed to trunk muscle fatigue resulting from the continuous contraction of deep trunk muscles in seated postures [9,10]. During prolonged sitting, the lumbar multifidus is passively stretched, resulting in increased co-contraction of the TrA and IO muscles to balance the back muscle forces. Consequently, the TrA and IO muscles become fatigued over time [9,10].

A significant increase in low back pain in the sitting condition was found in this study, suggesting that static loading of the lumbar spine during prolonged sitting may be associated with disc compression [54,55]. Healey et al. (2005) and Rodacki et al. (2003) proposed that persistent contraction of the superficial paraspinal muscles in patients with CLBP may produce greater compressive loading, increasing disc compression and reducing stature [25,27]. Moreover, the results of the current study demonstrate that the sitting condition reduced deep trunk muscle activation. These results may explain increased low back pain [14,56]. The results of the present study align with previous studies showing that perceived body discomfort increased significantly during prolonged sitting [56].

After the 5-week CSE with the ADIM technique, participants showed significantly improved stature changes during prolonged sitting. The stature change at the T_1_ (mean difference: 1.462 ± 1.752 mm; *p* < 0.001), T_2_ (mean difference: 1.756 ± 1.752 mm; *p* < 0.001), and T_3_ occasion (mean difference: 1.998 ± 2.653 mm; *p* < 0.001) was significantly improved before and after the 5-week CSE with the ADIM technique. Although our mean difference in stature change was only 1.99 mm, which did not reach the minimal clinically important stature change of 3 mm [12], there is evidence that changes in stature of a magnitude above 0.985 mm can be attributed to intervention effects in CLBP participants [57].

Thus, we show that the 5-week CSE with the ADIM technique can enhance recovery of disc height and reduce the loading on other spinal structures, which may facilitate a reduction in symptoms in patients with low back pain during prolonged sitting. This is consistent with the findings of Healey et al. (2005) and Lewis et al. (2014). They reported significant positive correlations between delayed stature recovery and higher levels of pain and disability [27,32].

Our results demonstrated that the 5-week CSE could improve trunk muscle endurance during prolonged sitting when compared with baseline. The results showed that both sides of TrA and IO significantly improved the MDF (at 0–10, 15–25th, and 28–38th min) when compared with baseline values. These revealed that muscle endurance increased after the 5-week CSE with the ADIM technique, which could be related to the specific effects of CSE with the ADIM technique on deep muscle activities. Macdonald et al. (2006) indicated that deep trunk muscles have a high percentage of type I muscle fibers, blood vessels, and mitochondria [58]. Therefore, CSE with the ADIM technique can improve the endurance of deep trunk muscles during prolonged sitting. Increased activity of the deep trunk muscle is thought to raise intra-abdominal pressure [59], resulting in decreased spinal loading [60,61]. This suggests that the CSE with the ADIM technique can increase deep trunk muscle activity in CLBP participants by reducing compression forces on the spine during prolonged sitting [8], leading to improved stature recovery.

The results of the five weeks of CSE with the ADIM technique significantly improved low back pain at all measurement time points (T_1_, T_2_, and T_3_) when compared with the baseline. However, the participants reported significantly increased pain intensity throughout the increased sitting time. Static loading of the lumbar spine increases stress in spinal structures [54,55]. These results may increase low back pain [14,56]. Thus, in addition to performing CSE with the ADIM technique, participants should perform the movement during the working day to prevent lower back pain during prolonged sitting.

The current study has some limitations. First, the investigation was conducted in a laboratory; the findings of this study may have limited ecological validity, and a real-life situation may be required in future investigations. Second, the participants were young, with a small age range (aged 21–29 years). Thus, the results might not be applicable to other age groups due to the variation in degenerative stage. Third, the current study was limited to the immediate effects of the 5-week CSE with the ADIM technique. Future studies should investigate the long-term follow-up effects of this program. Fourth, although the participants in this study reported a significant increase in low back pain during prolonged sitting, a history of previous injury did not meet our exclusion criteria. Therefore, low back pain may be due to other reasons besides prolonged sitting. Adding a history of previous injury in the exclusion criteria may better clarify the cause of low back pain in future studies. Lastly, the current study included only one group that performed the pre- and post-exercise interventions. Future investigations should include a control group or a comparison of CSE with other exercises to strengthen the findings.

## 5. Conclusions

This study demonstrated that a 5-week CSE with the ADIM technique affects the pain at rest and functional disability in sedentary workers with CLBP. Our result showed that CSE with the ADIM technique provides a protective effect on detrimental reductions in stature change and trunk muscle fatigue during prolonged sitting in young participants under controlled conditions in a laboratory. Based on these findings, we recommend that CLBP patients (aged 21–29 years) should perform CSE with the ADIM technique at home to reduce low back pain problems due to prolonged sitting activities.

## Figures and Tables

**Figure 1 ijerph-19-01904-f001:**
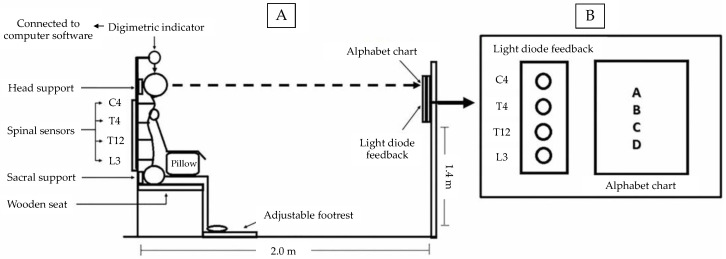
A seated stadiometer device: (**A**) participant position, (**B**) feedback chart. Source: [15].

**Figure 2 ijerph-19-01904-f002:**
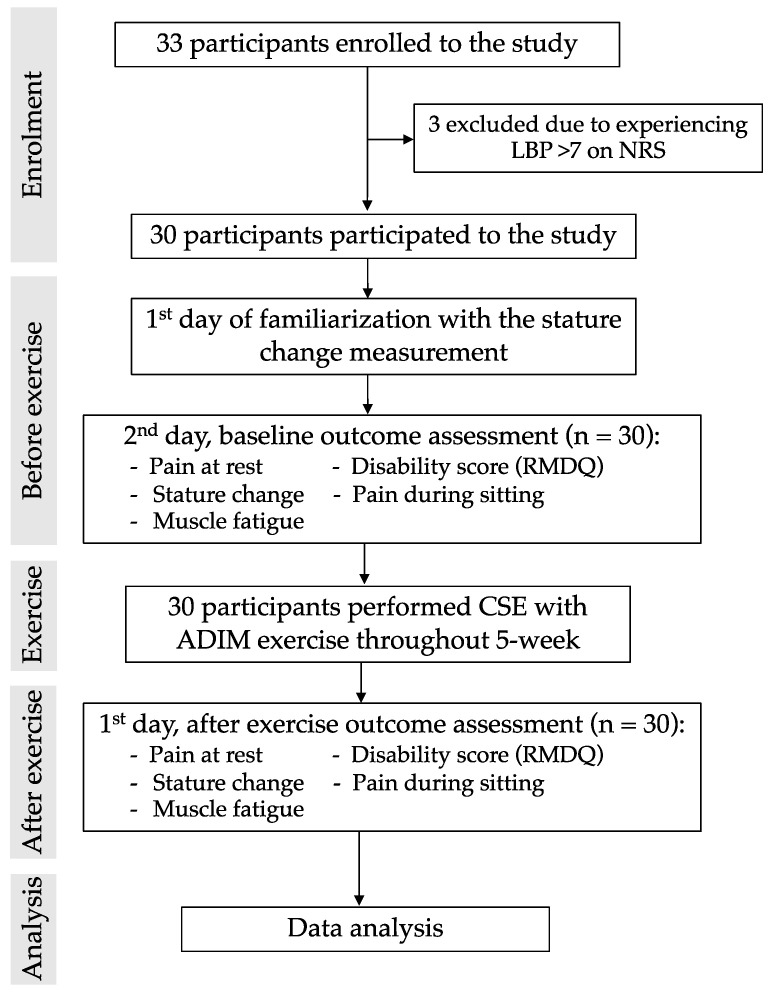
Flowchart of the study.

**Figure 3 ijerph-19-01904-f003:**
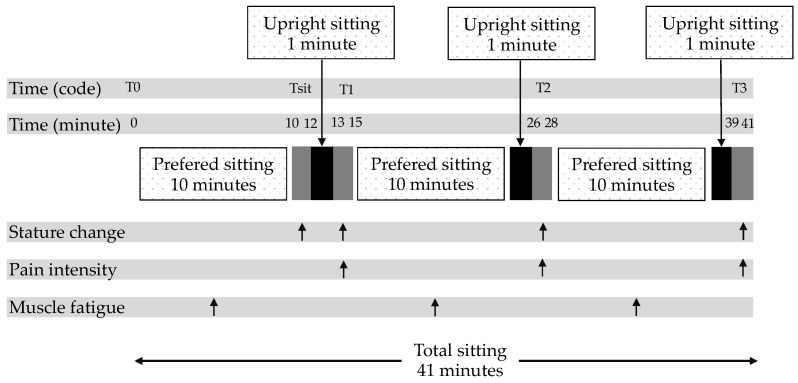
Overview of the prolonged sitting condition. The upright sitting periods (back zone) are presented below the axis. Arrows illustrate times of outcome measurement: stature change, pain intensity, and muscle fatigue. Participants sit in the seated stadiometer and perform the upright sitting three times (at 12–13, 25–26, and 38–39 min) throughout the prolonged sitting time of 41 min. Stature change measurements are collected at Tsit (10–12 min), T1 (13–15 min), T2 (26–28 min), T3 (39–41 min). Pain intensity measurements are collected at T1 (13–15 min), T2 (26–28 min), and T3 (39–41 min). Muscle fatigue measurements are collected at 0–10, 15–25, and 28–38 min.

**Figure 4 ijerph-19-01904-f004:**
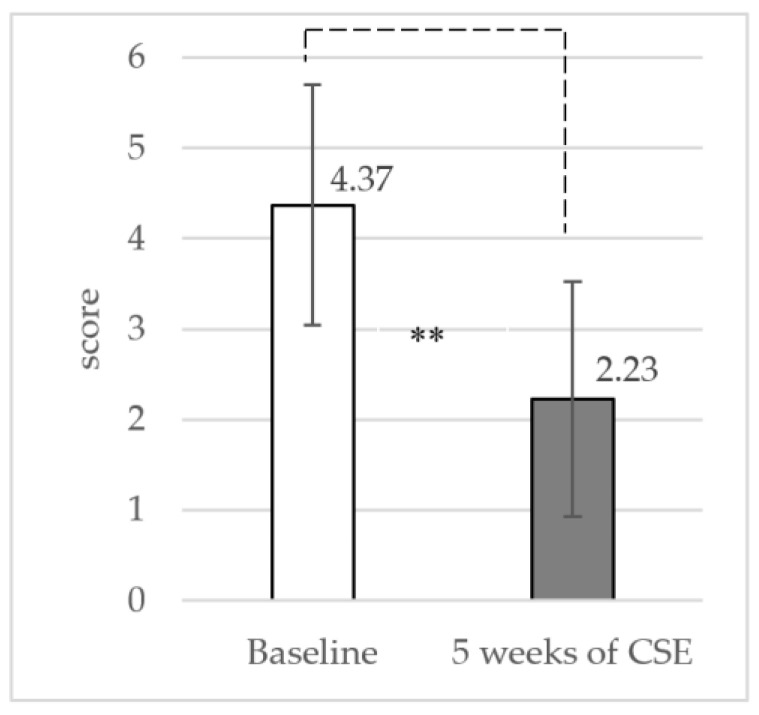
Mean and standard deviation (SD) of pain intensity from baseline to 5 weeks of CSE (** *p* < 0.001).

**Figure 5 ijerph-19-01904-f005:**
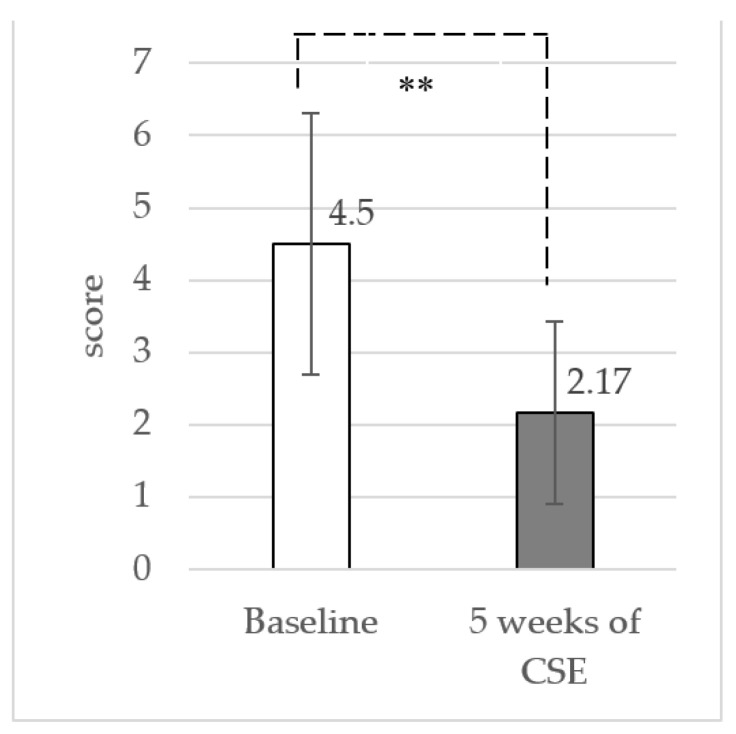
Mean and standard deviation (SD) of functional disability from baseline to 5 weeks of CSE (** *p* < 0.001).

**Table 1 ijerph-19-01904-t001:** Demographic characteristics of participants.

Characteristics	Male (n = 15)	Female (n = 15)	Total (n = 30)
Age (years), mean ± SD	25.67 ± 3.35	26.07 ± 3.37	25.87 ± 3.31
Weight (kg), mean ± SD	63.93 ± 7.94	52.80 ± 4.84	58.37 ± 8.59
BMI (kg m^−2^), mean ± SD	22.11 ± 1.90	20.95 ± 1.28	21.53 ± 1.70
Sitting height (cm), mean ± SD	87.93 ± 5.38	84.50 ± 3.39	86.22 ± 4.75
Standing height (cm), mean ± SD	169.80 ± 5.16	158.67 ± 4.70	164.23 ± 7.45
Smoking status	no	no	no
Occupation, n (%)			
-Student	12 (80)	13 (86.67)	25 (83.33)
-Office worker	3 (20)	2 (13.33)	5 (16.67)
Working time (hours/day), mean ± SD	8.60 ± 2.95	7.60 ± 2.13	8.10 ± 2.58
Period of LBP (month), mean ± SD	10.73 ± 6.18	10.53 ± 4.60	10.63 ± 5.35
Disability index score, mean ± SD	4.20 ± 1.82	4.80 ± 1.82	4.50 ± 1.81
Pain scale 24 h (score), mean ± SD	4.27 ± 1.33	4.47 ± 1.36	4.37 ± 1.33
Note: SD = Standard deviation; BMI = Body mass index.			

**Table 2 ijerph-19-01904-t002:** The stature change during sitting before and after CSE with ADIM.

	T_sit_ (mm) Mean ± SD (95%CI)	Mean Change from T_sit_ (mm) Mean ± SD (95%CI)		
		T_1_	T_2_	T_3_
Baseline	−4.266 ± 2.221 (−5.095 to −3.437)	−3.999 ± 1.482 (−4.553 to −3.446) ^g*h**^	−5.782 ± 1.605 (−6.382 to −5.183) ^f^^*h^^*^	−7.365 ± 2.180 (−8.179 to −6.552) ^f^^**g*^
After 5 weeks of CSE with ADIM technique	−3.864 ± 1.986 (−4.605 to −3.122)	−2.538 ± 1.004 (−2.913 to −2.163) ^g^^*h^^**^	−4.027 ± 1.306 (−4.515 to −3.539) ^f^^**h^^*^	−5.367 ± 1.258 (−5.837 to −4.897) ^f^^**g^^*^
*p*-value Between	0.458	**0.001**	**0.001**	**0.001**

Note: Data presented as Mean ± standard deviation (SD), T_sit_ = after sitting for 10 min, T_1_ = 13th–15th min, T_2_ = 26th–28th min, T_3_ = 39th–41st min, f = significant difference from T_1_, g = significant difference from T_2_, h = significant difference from T_3_ (* significant difference at *p*-value < 0.008, ** significant difference at *p*-value < 0.001).

**Table 3 ijerph-19-01904-t003:** Comparisons pain intensity during sitting before and after received CSE with ADIM.

Conditions	Times				*p*-Value within Conditions
	T_sit_	(T_1_)	(T_2_)	(T_3_)	
Baseline	3.00 (2.00–5.00) ^f^*^g^**^h^**	3.00 (3.00–5.00) ^e^*^g^*^h^*	3.50 (3.00–6.00) ^e^**^f^*	3.50 (3.00–6.00) ^e^**^f^*	0.001
After 5 weeks of CSE with ADIM technique	1.00 (0.00–2.00) ^g^*^h^**	1.00 (0.00–2.25) ^g^*^h^*	1.50 (0.00–3.00) ^e^*^h^*	2.00 (0.00–3.00) ^e^**^f^*^g^*	0.001
*p*-value Between	0.632	0.001	0.001	0.001	

Note: Data presented as Median (interquartile range), *p*-value from the Friedman test, ** significant difference at *p*-value < 0.001, * significant difference at *p*-Value < 0.05 by the Wilcoxon signed-rank test), T_sit_ = after sitting for 10 min, T_1_ = 13th–15th min, T_2_ = 26th–28th min, T_3_ = 39th–41st min, e = significant difference from T_sit_, f = significant difference from T_1_, g = significant difference from T_2_, h = significant difference from T_3_.

**Table 4 ijerph-19-01904-t004:** Comparison muscle fatigue between before and after performed CSM with ADIM exercise during prolong sitting.

Muscle Fatigue (Hz.)	Right		*p*-value	Left		*p*-Value
	Baseline Median (Interquartile Range)	After 5 Weeks of CSE Median (Interquartile Range)		Baseline Median (Interquartile Range)	After 5 Weeks of CSE Median (Interquartile Range)	
RA (0–10th min) (15th–25th min) (28th–38th min)	25.71 (24.99–27.25) 25.70 (917.29–27.64) 25.72 (25.70–28.69)	25.70 (25.69–25.71) 25.70 (23.91–25.71) 25.71 (25.69–25.72)	0.713 0.992 0.144	25.71 (15.72–28.95) 25.71 (16.95–27.34) 25.71 (25.69–28.70)	25.70 (25.12–25.71) 25.71 (25.69–25.72) 25.70 (24.72–25.71)	0.128 0.926 0.130
*p*-value	0.177	0.441		0.852	0.084	
TrA & IO (0–10th min) (15th–25th min) (28th–38th min)	42.59 (34.58–42.72) b*c* 35.99 (35.69–37.99) a* 35.71 (33.21–36.45) a*	46.71 (46.63–48.32) 45.30 (40.80–47.96) 45.70 (41.99–48.71)	0.001 ** 0.001 ** 0.001 **	42.70 (35.37–42.73) b*c* 36.15 (34.19–37.32) a* 35.95 (33.53–37.21) a*	46.71 (44.88–48.69) 45.45 (41.49–47.23) 45.99 (43.45–48.65)	0.001 ** 0.001 ** 0.001 **
*p*-value	0.001	0.058		0.001 *	0.503	
ICLT (0–10th min) (15th–25th min) (28th–38th min)	35.70 (33.21–36.03) 35.69 (33.28–35.72) 35.70 (33.85–35.71)	36.14 (35.69–37.94) 35.72 (35.30–37.33) 36.70 (33.85–35.71)	0.060 0.206 0.153	35.70 (34.81–35.72) 35.69 (33.66–36.21) 35.69 (33.58–35.71)	35.71 (34.39–37.76) 35.71 (35.66–36.47) 35.71 (35.69–37.05)	0.524 0.289 0.360
*p*-value	0.873	0.644		0.721	0.594	
LM (0–10th min) (15th–25th min) (28th–38th min)	49.04 (46.42–52.79) 49.41 (46.96–53.57) 49.21 (47.71–55.69)	52.65 (47.71–53.57) 52.74 (48.71–53.57) 52.34 (48.72–55.72)	0.185 0.098 0.082	49.36 (45.71–52.38) 49.21 (47.71–55.70) 49.84 (48.00–55.69)	51.90 (48.22–53.57) 52.33 (48.15–55.40) 52.42 (48.68–55.71)	0.175 0.544 0.237
*p*-value	0.695	0.341		0.273	0.125	

Note: Data presented as Median (interquartile range). Significant difference at * *p*-value < 0.05, ** *p*-value < 0.001 by Wilcoxon signed-rank test. a = significant difference from 0–10th, b = significant difference from 15th–25th, c = significant difference from 28th–38th.

## Data Availability

The data will be available for anyone who wishes to access them for any purpose and contract should be made via the corresponding author (rungthiprt@gmail.com).

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
