# Peer review of "The Effect of Core Stabilization Exercise with the Abdominal Drawing-in Maneuver Technique on Stature Change during Prolonged Sitting in Sedentary Workers with Chronic Low Back Pain"

_ijerph, 2022, doi:10.3390/ijerph19031904_

Round 1

Reviewer 1 Report

the paper “ The Effect of Core Stabilization Exercise with the Abdominal Drawing-in Maneuver Technique on Stature Change during Pro-longed Sitting in Sedentary Workers with Chronic Low Back Pain”

The study is interesting and addresses a problem, that of addressing the problem of LBP in the millions of workers who spend more and more hours sitting in working places in front of a screen, and uses a rational and well-documented work scheme.

First of all, the paucity of the number of tested subjects, almost all young students well motivated to perform the exercises. This sample does not allow the results to be directly transferred to a more general population. Second, it is necessary to perform evaluations in real life situations, as reported in lines 359-363.

I believe it is necessary to report more clearly these limitations. Therefore, these points should be underlined and reported both in the conclusions section and in the abstract of the article.

Author Response

Response to Reviewer 1 Comments

The authors appreciate the feedback from reviewer 2 for your precious time in reviewing our manuscript. We have responded to your comments on a point-by-point basis. The revised version of the manuscript is marked with yellow highlight changed.

Point 1: First of all, the paucity of the number of tested subjects, almost all young students well motivated to perform the exercises. This sample does not allow the results to be directly transferred to a more general population. Second, it is necessary to perform evaluations in real life situations, as reported in lines 359-363.

I believe it is necessary to report more clearly these limitations. Therefore, these points should be underlined and reported both in the conclusions section and in the abstract of the article.

 Response 1: Thank you for your suggestion. We have added more the detail about limitations of this study as follows:

“The current study has some limitations. First, the investigation was conducted in a laboratory; the findings of this study may have limited ecological validity, and a real-life situation may be required in future investigations. Second, the participants were young, with a small age range (aged 21–29 years). Thus, the results might not be applicable to other age groups due to the variation in degenerative stage.”

Section: discussion, Line: 385-389

We have also added the information about the generalizability of this study in conclusion and abstract section as follows:

 “The CSE with ADIM technique has been shown to provide a protective effect on detri-mental reductions in stature change and trunk muscle fatigue during prolonged sitting in young participants, under controlled conditions in a laboratory. This information may help to prevent the risk of LBP from prolonged sitting activities in real life situations.”

Section: abstract, Line: 27-31

 “Our result showed that CSE with the ADIM technique provides a protective effect on detrimental reductions in stature change and trunk muscle fatigue during prolonged sitting in young participants under controlled conditions in a laboratory. Based on these findings, we recommend that CLBP patients (aged 21–29 years) should perform CSE with the ADIM technique at home to reduce low back pain problems due to prolonged sitting activities.” 

Section: conclusion, Line: 401-406

Reviewer 2 Report

The study design and methods are reasonably written. There are some issues to be addressed.

  1. The major flaw in the study is the lack of control group. Is it a planned study design? or the authors are planning to do a full RCT and this is part of the pilot study? The authors should explain and provide a good justification of only performing this study with  1 group.
  2. The participants are instructed to perform the exercises on a daily basis. It is stated that they would record their exercises done in a logbook, so what is the compliance rate? 
  3. Who is the person collecting the pain score and functional disability data? Is there any blinding of asssessor? 
  4. what is the implication of a significant change in the stature measurement? what is the relationship to back pain? this may need to be addressed in Discussion
  5. Was there any long-term followup? how can you ensure that the training effect of the core exercises are maintained? 

Author Response

Response to Reviewer 2 Comments

The authors appreciate the feedback from reviewer 2 for your precious time in reviewing our manuscript. We have responded to your comments on a point-by-point basis. The revised version of the manuscript is marked with grey highlight changed.

Point 1: The major flaw in the study is the lack of control group. Is it a planned study design? or the authors are planning to do a full RCT and this is part of the pilot study? The authors should explain and provide a good justification of only performing this study with 1 group.

Response: Thank you for your question. The current study is planned study with 1 group design because we need to control individual factors that affect to stature change such as the morphology of the intervertebral disc. Our study was accorded to the study of Lewis et al. (2014). They also measured stature recovery of pre- and post- general exercise intervention in one group. Therefore, the design of this study is within-subject repeated-measures design.

However, we have considered for your comment and agreed to add this issue in our limitation section as follows:

“Lastly, the current study included only one group that performed the pre- and post-exercise interventions. Future investigations should include a control group or a comparison of CSE with other exercises to strengthen the findings.”

Section: discussion, Line: 396-398

Reference

- Lewis, S.; Holmes, P.; Woby, S.; Hindle, J.; Fowler, N. Changes in muscle activity and stature recovery after active rehabilitation for chronic low back pain. Man Ther 2014, 19(3), 178-83.   doi: 10.1016/j.math.2014.01.008.

Point 2: The participants are instructed to perform the exercises on a daily basis. It is stated that they would record their exercises done in a logbook, so what is the compliance rate?

Response: Thank you for your question. All the participants are record their exercises done in a logbook. The compliance rate is 100%. It is due to the researcher made a telephone call to motivate the participants every week to continue their daily home exercises. Moreover, all participants were re-assessed their exercise twice a week with the physical therapist as mentioned at the line 224-237 (section: 2.7. Core stabilization exercise (CSE) with ADIM technique).

Point 3: Who is the person collecting the pain score and functional disability data? Is there any blinding of asssessor? 

Response: Thank you for your question. As this study was a within-subject repeated-measures design it may be difficult to blind assessors. However, we have added the name of researchers who collecting each outcome measurement to clarify your question as follows:

“The process of stature change measurement was collected by Researcher PS.”

Section: outcome measurements, Line: 137

“Trunk muscle fatigue was collected by Researcher PS.”

Section: outcome measurements, Line: 150-151

“Pain intensity was assessed using an 11-point numerical rating scale (0–10 NRS). Subjective measures of pain were obtained from the NRS, employed to assess pain on a scale ranging from 0 (no pain) to 10 (worst possible pain) [31, 39]. This outcome measurement was evaluated by Researcher TC.”

Section: outcome measurements, Line: 153-156

“The Roland Morris disability questionnaire (RMDQ) Thai version was used to as-sess functional disability due to low back pain [40]. This questionnaire includes 24 items [41], which were rated by Researcher TC.”

Section: outcome measurements, Line: 160

“The exercise program was supervised by a physical therapist with 30 years of experience (RP).”

Section: outcome measurements, Line: 222-223

Point 4: what is the implication of a significant change in the stature measurement? what is the relationship to back pain? this may need to be addressed in Discussion

Response: Thank you for your question. We have added the detail about the implication of a significant change of stature and the relationship to back pain as follows:

“After the 5-week CSE with the ADIM technique, participants showed significantly improved stature changes during prolonged sitting. The stature change at T1 (mean difference: 1.462 ± 1.752 mm; p < 0.001), T2 (mean difference: 1.756 ± 1.752 mm; p < 0.001), and T3 occasion (mean difference: 1.998 ± 2.653 mm; p < 0.001) were significantly im-proved before and after the 5-week CSE with the ADIM technique. Although our mean difference in stature change was only 1.99 mm, which did not reach the minimal clinically important stature change of 3 mm [12], there is evidence that changes in stature of magnitude above 0.985 mm can be attributed to intervention effects in CLBP participants [57].

Thus, we show that the 5-week CSE with the ADIM technique can enhance recovery of disc height and reduce the loading on other spinal structures, which may facilitate a reduction of symptoms in patients with low back pain during prolonged sitting. This is consistent with the findings of Healey et al. (2005) and Lewis et al. (2014). They reported significant positive correlations between delayed stature recovery and higher levels of pain and disability [27, 32].”

Section: discussion, Line: 351-364

Point 5: Was there any long-term followup? how can you ensure that the training effect of the core exercises are maintained?

Response: We have added a limitation about long-term follow-up as follows:

“Third, the current study was limited to the immediate effects of the 5-week CSE with the ADIM technique. Future studies should investigate the long-term follow-up effects of this program.”

Section: discussion, Line: 389-391

Reviewer 3 Report

Reviews

  1. Abstract:
  • Line 14-15 I recommend writing “compensate for the forces originating from the upper body”
  • I also recommend providing the instrumentation and sensor technology in abstract. It is important to provide what measurement technology/sensors you used which leads to your conclusion?

  1. Introduction
  • Line 55-56, Suggest writing “superficial trunk muscle increased activation……….”
  • I also suggest citing the following literature about neural control and muscle activation (trunk) for compensation (Singh, R. E., Iqbal, K., White, G and Hutchinson, T. E., 2018).
  • Overall, the introduction is written nicely. The novelty and aim of research are quite clear. I can read and understand from the introduction that previous studies have not investigated long term effect of ADIM, but your study is examining long term impact of ADIM and its effect on LBP etc. However, I suggest presenting about ADIM technique in a paragraph in introduction.

  1. Methods

  • Line 123-124, To control the spinal alignment, sensor data was used as a measure. Its appropriate to write this statement in such way as sensors do not control rather, they are used as a measure for control.
  • I am curious to know whether participants were on any pain killers and/or other medication during this study. Kindly provide details.
  • Also, the patient recruited in the study were having LBP due to long duration of sitting. Kindly provide information whether it was long-term sitting that was the sole reason for LBP or they have had some injury in the past and with long-term sitting it contributed to LBP.
  • Figure 3 caption should be clearer. The grey and black bands what do they represent, What does those arrows mean? It is not clear; I believe they are time duration. I suggest more description for figure 3 in caption. I suggest reading nature medicine paper. They are quite extensive and clear about presenting and describing their images in figure caption.

Author Response

Response to Reviewer 3 Comments

The authors appreciate the feedback from reviewer 3 for your precious time in reviewing our manuscript. We have responded to your comments on a point-by-point basis. The revised version of the manuscript is marked with green highlight changed.

Point 1: Abstract, Line 14-15 I recommend writing “compensate for the forces originating from the upper body”

Response:  Thank you for your suggestion. We have rewritten this sentence as follows:

“To enhance stature recovery, lumbar spine stabilization by stimulating the deep trunk muscle activation for compensation forces originating from the upper body was introduced.”

Section: abstract, Line: 14-16

Point 2: I also recommend providing the instrumentation and sensor technology in abstract. It is important to provide what measurement technology/sensors you used which leads to your conclusion?

Response: Thank you for your comment. We have tried to add this issue at our abstract as follows:

“Statue change was measured using a seated stadiometer with a resolution of ±0.006 mm.”

Section: abstract, Line: 22-23

Point 3: Introduction, Line 55-56, Suggest writing “superficial trunk muscle increased activation……….”

Response: Thank you for your suggestion. We have rewritten this sentence as follows:

“Increased superficial trunk muscle activation occurs to compensate for deep trunk mus-cle dysfunction [24, 25], in which the neural control subsystem attempts to maintain spinal stability [18, 26]. Increased activation of the superficial trunk muscle can com-press the spinal structure and lead to delayed stature recovery [25, 27].”

Section: introduction, Line: 58-62

Point 4: I also suggest citing the following literature about neural control and muscle activation (trunk) for compensation (Singh, R. E., Iqbal, K., White, G and Hutchinson, T. E., 2018).

Response:  Thank you for your suggestion. We have cited the reference form your suggestion as follows: 

“Increased superficial trunk muscle activation occurs to compensate for deep trunk mus-cle dysfunction [24, 25], in which the neural control subsystem attempts to maintain spinal stability [18, 26].”

Section: introduction, Line: 58-61

Reference:

  1. Singh, RE; Iqbal, K; White, G; Hutchinson, TE. A Systematic Review on Muscle Synergies: From Building Blocks of Motor Behavior to a Neurorehabilitation Tool. Appl Bionics Biomech 2018, 22, 3615368. doi: 10.1155/2018/3615368.

Point 5: Overall, the introduction is written nicely. The novelty and aim of research are quite clear. I can read and understand from the introduction that previous studies have not investigated long term effect of ADIM, but your study is examining long term impact of ADIM and its effect on LBP etc. However, I suggest presenting about ADIM technique in a paragraph in introduction.

Response: Thank you for your suggestion. We have added the detail about ADIM technique in introduction section as follows:

“The ADIM technique is known to elicit preferential recruitment of the transversus abdominis muscle, with minimal activation of the superficial trunk muscles. For this technique, participants are instructed to ‘gently draw in their lower abdomen toward the spine’ [13].”

Section: introduction, Line: 65-68

Point 6: Methods, Line 123-124, To control the spinal alignment, sensor data was used as a measure. Its appropriate to write this statement in such way as sensors do not control rather, they are used as a measure for control.

Response: Thank you for your suggestion. We have rewritten the sentence as follows:

“These sensors were used as a measure for control to ensure that the participants maintained the same posture throughout the experiment period.”

Section: materials and methods, Line: 135-137

Point 7: I am curious to know whether participants were on any pain killers and/or other medication during this study. Kindly provide details.

Response: The participant reported, they did not use any drug for reducing their low back pain. We also added this sentence into our result section as follows:

“The participants reported that they did not use any drug for reducing their low back pain and had no advert of the exercise throughout the 5 weeks of the training.”

Section: result, Line: 259-261

Point 8: Also, the patient recruited in the study were having LBP due to long duration of sitting. Kindly provide information whether it was long-term sitting that was the sole reason for LBP or they have had some injury in the past and with long-term sitting it contributed to LBP.

Response: Thank you for your question. The current study did not exclude about history of previous injury, so we cannot actually know that participant’s low back pain was due to prolong sitting or previous injury. However, as Table 3, the result of our study reported a significant increase in low back pain during prolonged sitting condition.

We have agreed of your concern, so we have added this in our limitation section as follows:

“Fourth, although the participants in this study reported a significant increase in low back pain during prolonged sitting, a history of previous injury did not meet our ex-clusion criteria. Therefore, low back pain may be due to other reasons besides pro-longed sitting. Adding a history of previous injury as the exclusion criteria may better clarify the cause of low back pain in future studies.”

Section: discussion, Line: 391-396

Point 9: Figure 3 caption should be clearer. The grey and black bands what do they represent, What does those arrows mean? It is not clear; I believe they are time duration. I suggest more description for figure 3 in caption. I suggest reading nature medicine paper. They are quite extensive and clear about presenting and describing their images in figure caption.

Response: Thank you for your suggestion. We have rewritten caption of Figure 3 as follows:

“Overview of the prolonged sitting condition. The upright sitting periods (back zone) are presented below the axis. Arrows illustrate times of outcome measurement: stature change, pain intensity and muscle fatigue. Participants sit in the seated stadiometer and perform the upright sitting three times (at 12–13, 25–26 and 38–39 min) throughout the prolonged sitting time of 41min. Stature change measurements are collected at Tsit (10-12 min), T1 (13–15 min), T2 (26–28 min), T3 (39–41 min). Pain intensity measurements are collected at T1 (13–15 min), T2 (26–28 min) and T3 (39–41 min). Muscle fatigue measurements are collected at 0–10, 15–25 and 28–38 min.”

Section: materials and methods, Line: 202-209

Round 2

Reviewer 2 Report

The revised manuscript is much improved. I have no further comments